# Machine Learning Approach to Predict the Performance of a Stratified Thermal Energy Storage Tank at a District Cooling Plant Using Sensor Data

**DOI:** 10.3390/s22197687

**Published:** 2022-10-10

**Authors:** Afzal Ahmed Soomro, Ainul Akmar Mokhtar, Waleligne Molla Salilew, Zainal Ambri Abdul Karim, Aijaz Abbasi, Najeebullah Lashari, Syed Muslim Jameel

**Affiliations:** 1Department of Mechanical Engineering, Universiti Teknologi Petronas, Seri Iskandar 32610, Perak Darul Ridzuan, Malaysia; 2Mechanical Engineering Department, Quaid-e-Awam University of Engineering Science and Technology, Nawabshah 67450, Pakistan; 3Department of Petroleum and Gas Engineering, Dawood University of Engineering & Technology, M.A. Jinnah Road, Karachi 74800, Pakistan; 4School of Engineering, University of Galway, University Road, H91 TK33 Galway, Ireland

**Keywords:** temperature distribution, thermal energy storage, thermocline thickness, artificial neural networks, support vector machine, k-nearest neighbor district colling, temperature sensors

## Abstract

In the energy management of district cooling plants, the thermal energy storage tank is critical. As a result, it is essential to keep track of TES results. The performance of the TES has been measured using a variety of methodologies, both numerical and analytical. In this study, the performance of the TES tank in terms of thermocline thickness is predicted using an artificial neural network, support vector machine, and k-nearest neighbor, which has remained unexplored. One year of data was collected from a district cooling plant. Fourteen sensors were used to measure the temperature at different points. With engineering judgement, 263 rows of data were selected and used to develop the prediction models. A total of 70% of the data were used for training, whereas 30% were used for testing. K-fold cross-validation were used. Sensor temperature data was used as the model input, whereas thermocline thickness was used as the model output. The data were normalized, and in addition to this, moving average filter and median filter data smoothing techniques were applied while developing KNN and SVM prediction models to carry out a comparison. The hyperparameters for the three machine learning models were chosen at optimal condition, and the trial-and-error method was used to select the best hyperparameter value: based on this, the optimum architecture of ANN was 14-10-1, which gives the maximum R-Squared value, i.e., 0.9, and minimum mean square error. Finally, the prediction accuracy of three different techniques and results were compared, and the accuracy of ANN is 0.92%, SVM is 89%, and KNN is 96.3%, concluding that KNN has better performance than others.

## 1. Introduction

From the standpoint of heating and air conditioning, stratified thermal energy storage (TES) has become increasingly popular in recent years [1,2]. The advantage of adopting TES for comfort and cooling applications is that energy and cost concerns are no longer an issue. The actual performance of TES is determined by several elements, including mixing at the tank’s inflow, mixing of hot and cold water, heat loss to the environment, and aspect ratio [3].

The above-mentioned factors impact the temperature distribution in the TES tank during the charging period; the transition between hot and cold water in the tank is known as the thermocline thickness ($W_{TC}$). The performance of TES tank is determined through thermocline thickness ($W_{TC}$) [4,5,6]. Multiple methods have been proposed to calculate $W_{TC}$, such as a small-scale experimental setup, finite element analysis, computational fluid dynamics, and curve-fitting from sensor data.

Yoo et al. computed $W_{TC}$ by extrapolating thermocline edges from the thermocline’s mid-point. The region fringed to the linear slope of the thermocline profile is determined using interpolation. The thermocline edges were not determined at the true upper and lower limits of thermocline profiles, which is one of the methodology’s disadvantages [7]. Steward [8] conducted the steady-state (solving partial differential equations) model for stratified TES. Musser and Behnfleth [9] analyzed the performance of TES tank using the numerical method. Musser and Bahnfleth proposed a more reliable and simple technique of binding the thermocline zone by employing the dimensionless cut-off temperature on each edge of the thermocline region. The quantity of thermocline detected was suggested to be large enough to eliminate the impacts of tiny temperature changes at the thermocline’s extremities, but small enough to capture the majority of the temperature changes. This is how the dimensionless cut-off temperature is defined [9]. Unfortunately, the methods discussed here have two major drawbacks; first, when the temperature readings available are in discrete form, these methods cannot predict $W_{TC}$ accurately [10], and secondly, the computational complexity requires extensive physical-based knowledge. To cope with the above-mentioned problems, data driven approaches, including artificial neural network (ANN), support vector machine (SVM), and k-nearest neighbor (KNN), have been used in this study to predict thermocline thickness. Because of its learning ability and versatile mapping skills without requiring considerable physical based knowledge, [11] machine learning has attracted increased interest in the system modelling and simulation industry [12,13]. Machine learning (ML) have been used to mimic a variety of engineering applications (such as building energy systems, including TES [14,15], thermal performance in battery systems [16], human thermal comfort in passenger vehicles using an organic phase change material [17], agriculture [18], maintenance of gas pipelines subjected to corrosion [19], subsea pipelines annotation [20], predicting water saturation [21], viscosity of methane gas at high temperature conditions [22], integrity of corroded oil and gas pipelines [23]) other than machine learning-based indoor occupancy predictions on PCMs integrated building energy systems [24,25]. A state-of-the art review on the applications of the phase change materials on storage systems can be found in [26]. Machine learning applications in surrogate model and model predictive control are explored in [27]. A computational-efficient model was developed to predict the renewable generation in [28]. A supervised learning surrogate model was developed to improve prediction efficiency in [29].

Some researchers have focused on utilizing FFNN to forecast the TES tank’s temperature distribution. A water thermocline storage tank’s thermal stratification was modelled by G’eczy-Víg and Farkas [30] for both load and load-free conditions. The temperatures at different vertical positions were forecasted every 5 min, utilizing 12 inputs, including the outputs at a previous time-step. Additionally, [31] covered how ANN time-step affected stratification modelling. Soomro and Mokhtar [32] created an ANN model to forecast the variables of the sigmoid dose-response function describing the water TES tank’s temperature profile. Diez et al. [33] developed a model to predict the evolution of temperature distribution of a stratified solar water tank in static mode, in place of the dynamic processes of charging and discharging. The last time-matching step’s temperature values were used as inputs to predict the temperatures at various vertical layers at 10-min intervals. Recently, ANN has been effectively employed for the best control of a district cooling plant with a thermal energy storage tank for predicting the performance of the stratified chilled water TES of a heat pump system [34]. This paper suggests a novel method for controlling thermal energy storage (i.e., ice storage) in a district cooling system and efficiently predicting performance. Jia, Liu et al. 2022 created a system for TES operating strategy optimization by fusing physics-based modelling with deep learning [35]. The ANN has recently seen significant use in applications related to the energy system, as the aforementioned literature on ANN demonstrates. Based on the above discussion, it is clear from the literature that machine learning has recently drawn the attention of various fields and also in energy systems, but has not yet been well utilized for stratified thermal energy storage performance in district cooling plants; this is why, in this study, machine learning has been applied to predict the performance of a stratified thermal energy storage tank with sensor data. An algorithm’s success is determined by the nature of the problem, such as its variables, boundaries, and additional complications, such as patterns in data. The ability of an algorithm to solve a particular problem, however, does not guarantee that it will perform better than a random search [36,37]. Three machine learning techniques, artificial neural networks (ANN), support vector machines (SVM), and K-Nearest Neighbor (KNN), were used in this study.

The structure of the paper is as follows: in Section 2, the methods used to achieve the study objective are discussed. In Section 3, the results generated using machine learning algorithms are presented and discussed. Finally, Section 4 exhibits the conclusion and the future works.

## 2. Materials and Methods

### 2.1. Data Collection

The data in this study was collected from the operating TES tank installed at the UTP GDC plant. Figure 1 shows the layout of TES system at GDC UTP and Figure 2 represents the flow during the charging process of TES tank at the GDC plant.

After collecting the data from the TES sensors, the data was analyzed in an Excel worksheet. Figure 3 shows that during charging, the temperature distribution moves from the bottom to the top direction, and the width of the thermocline thickness increases with the passage of time. This behavior has been verified in many previous studies, such as [6,38,39,40]. Evolution of temperature distribution and thermocline thickness during charging is shown in Figure 3.

### 2.2. Description of TES System

The TES capacity is 10,000 RTh, connected with four vapor compression or electric chillers (EC) which charge the TES tank and have a capacity of 325 RTh each. The cooling demand of the UTP campus and mosque is fulfilled by the GDC plant. The 14 temperature sensors are installed vertically at a distance of 1 m in the tank, and the data acquisition system is used to collect the temperature data, as shown in Figure 1. During off-peak hours, the chillers are used to charge the tank, and during on-peak periods the tank is discharged to supply the chilled water to the campus, as shown in Figure 2. The tank is cylindrical, having an inside diameter of 22.3 m and a height of 15 m. The total storage capacity of the tank is 54,000 $m^{3}$. The lower nozzle is made up of a pipe with a diameter of 0.5 m located at a height of 1.82 m, and the upper nozzle has a diameter of 0.3 and its location is at 12.3 m; diffusers are attached with both nozzles. The insulation material used is polystyrene, with 300 mm thickness, and epoxy paint is used for the internal coating of the tank. The mass flow rate of the electric chiller is designed at 131 $m^{3}/h$, the inlet and outlet temperature are set at 13 °C and 6 °C, respectively.

The input to the machine learning models is the temperature distribution collected from the sensors installed vertically in the tank, and the output data for the machine learning models model is $W_{TC}$. The outputs were obtained using the non-linear regression technique proposed by [41,42]. Equation (1) has been used to predict $W_{TC}$.
(1)$$W_{TC}=\frac{2\times Log\times \left(\frac{1}{\theta }-1\right)}{S}$$
where
(2)$$\theta =\frac{\left(T-T_{c}\right)}{\left(T_{h}-T_{c}\right)}$$

### 2.3. Data Preprocessing

In order to have an accurate prediction model, dealing with the input data before feeding it to the algorithm is always desirable. Hence, in this study, the input data were normalized and smoothed with two different smoothing techniques, such as moving average filter and median filter. The procedure is shown in Figure 4 below.

A.Data Normalization

In this study, the data normalization was carried out using the minimax technique, using Equation (3). In this method, the rescaling of the outputs was carried out by transferring one range of values to another range of values [43,44]. Mostly the rescaling is carried out between [0, 1] or [−1, 1]. The linear interpretation formula, such as Equation (3), was used for the rescaling. In this study, the range of normalization was between 0 and 1. The normalization procedure could lessen the learning algorithm’s ambiguity regarding the significance of each parameter with a smaller amplitude. Figure 5 represents that how the data is normalized.
(3)$$X_{norm}=\frac{X-X_{min}}{X_{max}-X_{min}}$$
where *X* is the original value, $X_{min}$ is the minimum value and $X_{max}$ is the maximum value.

B.Data smoothing/denoising techniques

In this study, the moving average filter and median filter were used to denoise the data. The mean average filter replaces the point with the average values found for the number over a specified frame length, where the moving median filter works with the principle of processing the signal entry by entry, and replacing each input with median values over a certain frame length [45].

The moving average filter is expressed as:
(4)$$y[i]=\frac{1}{M}\overset{M-1}{\underset{j=0}{\sum }}x[i-j]$$
where the input values are x[i − j], the output value is y[j], and the frame duration is M.

The filter or the smoothing using moving average filter is expressed as:y = movmean(A,[m 0])(5)
where A is an array of input data, Y is smoothed data, and m is the frame length.

The filter or the smoothing using moving median filter is expressed as:y = movmedian(A,[m 0])(6)
where A is an array of input data, Y is smoothed data, and m is the frame length.

Figure 6 shows the behavior of the input and output data. The data clearly shows that there are some noises which must be denoised.

Hence the data was smoothed using the moving average filter and median average filter, as shown in Figure 7. The graph shows that T1, T2, T3, and T4 were more smoothed by the mean average filter, T5, T6, T7, T8, T9, and T10 were more smoothed by moving median filter, whereas T11, T12, T13, and T14 show the same smoothing magnitude for both smoothing techniques at the same frame length. The frame length for both filters was 20.

Figure 8 shows smoothed and normalized versus unsmoothed and normalized data. The blue color graph is for unsmoothed normalized data, the red color graph for mean average filter-smoothed and normalized data, and the yellow one is for median average filter-smoothed and normalized data.

### 2.4. Data Division

Once the dataset was obtained from the plant, it was stored in the CSV file, then the file was uploaded in the MATLAB toolbox where it was divided into three parts. A subset of 185 (70%) from the whole data was used for training purposes. A subset of 39 (15%) from the whole dataset was used for cross validation. A subset of 39 (15%) was used for testing purposes. Table 1 shows the example of division of dataset.

### 2.5. ANN Modeling

After the data collection and preprocessing, ANN modeling was performed using the feed-forward backpropagation technique [44] as shown in Figure 9. The mathematical form of artificial neural network is presented in below.

#### Steps in ANN Modeling

Step 1 Determine the output of the input layer.

Step 2 Determine the output of the hidden layer.

The corresponding normalized input value is multiplied with the corresponding initial weight and these products are added together. This summation is again added to with the variable bias. The resulting sum is applied to the activation function. The activation function employed is the log sigmoid function, due to its best prediction capability as proved in some studies, including but not limited to [46], as shown in Equation (7).
(7)$$Output=\frac{1}{1+e^{-x}}$$
where x is the result of adding the inputs and weights.

Step 3 Determine the output of the output layer.

The same logic, involving multiplication of weights, with inputs followed by summation and its application to the activation function (as in Step 2) is applied, but the activation function used here is linear function, as shown in the Equation (8).
(8)Output=x

Step 4 Determine the error of the model.

For each training pattern, the error of the output layer is compared to the desired output, and the difference between the two is the error, which is determined using Equation (9).
(9)E=T−Y

*Y* is the forecast value, while T is the target value.

Step 5 Calculate the square of the error.

The square of the error can be calculated using Equation (10).
(10)$$E^{2}={\left(T-Y\right)}^{2}$$

A.Backward Pass

The purpose of the backward pass is to calculate errors in the hidden and input layers and modify the weights for better predictions.

Step 1 Determine the node’s error in the hidden layer.

The derivative of the activation function is multiplied by error, and the result is used to determine the error. The activation function is derived as follows:(11)$$\delta_{2}=Y\times \left(1-Y\right)\times \left(Y-T\right)$$

The activation function’s derivative is given by $Y\times \left(1-Y\right)$

Step 2 Determine the node’s error in the input layer.

The error is determined by multiplying the activation function’s derivation by the error.
(12)$$\delta_{1}=Y\times \left(1-Y\right)\times \left(Y-T\right)$$

Step 3 Adjustment of the weights.

There are several methods for finding the minima of a parabola or any function in any dimension. One of the best techniques for training ANNs is the Levenberg–Marquardt optimization algorithm, particularly when the number of weights is high. Although requiring more memory than other techniques, this method is strongly advised as the first choice for supervised tasks. Equation (13) is used to update the weights in a Levenberg–Marquardt training function (LMTF).
(13)$$w_{i+1}=w_{i}-{\left(J_{i}^{T}J_{i}+\mu_{i}I\right)}^{-1}J_{i}^{T}e_{i}$$

Step 4 Determine whether or not the training should continue.

Repeat the previous steps until the minimum error is reached or the set number of iterations has been exceeded.

Step 5 Denormalization of the output.

This step is crucial to return the values to their original, unnormalized form using Equation (14).
(14)$$X=X_{norm}\times \left(X_{max}-X_{min}\right)+X_{min}$$

### 2.6. Support Vector Machine

The supervised method (the most-isolation approach) is the support vector machines method. By supplying numerical data, it solves classification and regression problems [47]. The precise location for the hyperplane is determined by the support vector machine process through optimization until the hyperplanes are positioned in the maximum margin [48]. The hyperplane with the highest margin is capable of accurately classifying data or other input parameters [49]. The key concern in support vector machine (SVM) is how to determine the hyperplane’s ideal position. The most acceptable position should be chosen after conducting many tries. The hyperplane’s main function is to divide the two classes. Thus, widening the decision boundary between the two classes is suitable. The line crossing across the support vector is referred to as the separate lines to indicate the decision boundary. The concept of SVM is depicted in Figure 10.

All the input data is classified into the necessary classes using a support vector machine. The label, *y_i_*, and the set of data, “*x*1… *x*n,” are required. The input data is either predicted to be in the class label, *y_i_* = 1 or in the class label, *y_i_* = −1 [50]. Accordingly, it is written as:(15)$$\mathrm{Data}={\left\{\left(x_{i},y_{i}\right)|x_{i}\in {ℜ}^{p},y_{i}\in \left(-1,+1\right)\right\}}_{i=1}^{n}$$
where *n* is the number of data points and *p* is the feature dimension.

The hyperplane equation is expressed as:(16)w.x=0

The equation for 2D space is as follows: (17)Y=wx+b

From Equations (1)–(3), we can generalize that:(18)$$\begin{array}{l} W\cdot \overset{\bar{}}{X_{i}}+b\geq 0\forall i:y_{i}=+1.\\ W\cdot \overset{\bar{}}{X_{i}}+b\leq 0\forall i:y_{i}=-1. \end{array}$$
where $W=\left(w1\ldots wd\right)$ is the dimensions row vector corresponding to the normal axis to the classifier, b is the bias, and Xi is a provided row vector associated with the ith data point. $yi\in \left\{-1,+1\right\}$ is a binary class variable of the ith data point. The bias b controls the distance of the classifier line from the origin as the vector W controls the classifier line’s position.

### 2.7. K-Nearest Neighbor

The fundamental supervised algorithm is commonly referred to as k-NN. The concept behind KNN is used in many other machine learning methods, therefore to understand other techniques, studying KNN could be a great place to start [51]. The instances are not stored throughout the training phase. In training, some reasonable indexing is required to quickly identify the classes. The k-nearest classes of the one instance must be found via KNN [52]. The effectiveness of classification is dependent on the preference of similarity by calculating the distance between two instances. The k-nearest neighbor algorithm is shown in Figure 11. The Euclidean distance method is the most practical of the various ways to determine the distance between two points. The Euclidean distance between two points in XY dimensions is calculated as follows:(19)$$\begin{array}{c} \mathrm{Euclidean}\;\mathrm{Distance}\;\mathrm{between}\;\mathrm{point}\;A_{1}\;\mathrm{and}\;B_{2}=\\ \sqrt{{\left(X_{2}-X_{1}\right)}^{2}+{\left(Y_{2}-Y_{1}\right)}^{2}} \end{array}$$

## 3. Development of Prediction Model

### 3.1. Classifier Tuning and Validation Techniques

The performance of the classification model prediction is tested using a variety of techniques. It is conventional to use 70% of the data set for training and the remaining 30% for testing the model’s efficacy. This approach is known as leave-p-out. Cross-validation is the second type of model training and testing strategy, and works by dividing the given dataset into training and validation data subsets and then rotating the evaluations among the subsets; it can assess a classifier’s generalized overall accuracy on an unbiased dataset. When the data set is small, cross validation is used. A generalization called k-fold cross-validation divides the data into k−1 training sets and a validation set. The partitions must roughly be equal in size and are formed by randomly choosing data. The average of each performance metric, such as classification accuracy, will be the outcome of cross-validation. K-fold cross validation means using the partition just once while repeating the validation k times. MATLAB programming language was used to develop the classification algorithms. During development of the SVM and KNN prediction model, the frame length for both moving average filter and moving median filter was 20. The frame length was varied and finally a frame length which leads to the best classification accuracy was selected.

### 3.2. Hyper Parameter Optimization for the Model

The machine learning models are highly dependent on the following hyerparameters as shown in the Table 2.

### 3.3. Evaluation Performance Indices

After the prediction was achieved from the machine learning models, the output of machine learning models was compared with the actual output, i.e., the training dataset. To compare the performance of the ANN and non-linear regression model results, MSE, as shown in Equation (20), and R^2^, as shown in Equation (21), were used in this study as follows. However, for the support vector machine and KNN, the confusion matrix [53] was generated, which shows the prediction accuracy.
(20)$$MSE=\frac{1}{n}\overset{n}{\underset{n=1}{\sum }}{\left(T-O\right)}^{2}$$
(21)$$R^{2}=1-\frac{\sum_{i=1}^{n}{\left(T-O\right)}^{2}}{\sum_{i=1}^{n}{\left(T-\bar{O}\right)}^{2}}$$

## 4. Result and Discussion

This section has been divided into two parts. The first part consists of the results of the ANN architecture-building and the second part is related to the validation or comparison of the output values with the real behavior of the thermocline thickness. The algorithms are assessed using the R-square and MSE metrics that were previously discussed. For the train and test datasets, the predicted and actual values are compared. The projected and experimental data in the best-case scenario should be a 45° slope line that may be shown as the best fit curve to visually analyze the deviation.

### 4.1. Prediction with ANN Model Architecture

The optimum ANN setup, which is dependent on control parameters such as the number of hidden neurons, the number of hidden layers, and the learning rate, was achieved. Back-propagation with the LM algorithm was the learning algorithm used here, as indicated in Table 3.

A total of 263 training data sets were used to train the network. The impact of the number of neurons on model performance can be seen in Figure 12 in terms of R-square and MSE, respectively. The number of neurons has an impact on the R-square of the model, with 10 neurons achieving the best results, and after that, the model began to overtrain, as experienced in [54]. The impact of the learning rate on the R-square and MSE of the model is also shown in Figure 12. The minimum value of MSE and maximum value of R-square is noted at 0.01 learning rate.

### 4.2. Prediction of Thermocline Thickness

Figure 13 shows the overall regression of the collected and predicted data and Figure 14 shows the comparison between the ANN predicted and actual values during the charging period. The graph shows a very motivated representation, as the data is very close to the central line and gives an R-square of 0.92 for $W_{TC}$. According to the theory, $W_{TC}$ increases during charging time due to mixing and conduction effects. It can be seen from Figure 14 that during the charging period the $W_{TC}$ increased from 1.94 to 3.5, which supports the theoretical behavior of thermocline during charging. The average error between the ANN predicted and actual values is 2.4%. Figure 14 shows the comparison between the calculated and ANN predicted values of $W_{TC}$; it can be seen that the $W_{TC}$ has increased with respect to charging time as expected [9,41,55,56].

Figure 15 represents the changes in the MSE for training, validation, and testing; with respect to the iterations, it can be seen from the graph that the best result was achieved at the 23 iterations, which means that the minimum value of MSE is achieved at 23 iterations or epochs.

The outputs of the model and the actual values are compared for training, validation, and testing and the results are shown in Figure 16. The R value shows the closeness to the targeted values; if the value is greater than 0.9 it means the predictions are satisfactory, and it can be seen from the Figure 16 that the R values for training, validation, and testing are 0.97, 0.94, and 0.95, respectively. A total number of 263 data points were used in the comparison. The closeness of the data to the central line indicates the accuracy of the prediction of the model. It can be seen that the data is very close to the trend line, showing a very good prediction result. The R-square for $W_{TC}$ comparison is 0.92.

### 4.3. Prediction via KNN and SVM

The Input data magnitude with it’s label shown in Table 4 below.

(A).
*Prediction via KNN*


Figure 17 shows the confusion matrix for k-nearest neighbor with mean average filter prediction, and it shows that the maximum false positive value is 1, which happened in class 7; the maximum false negative is 1, which happened in class 2, 10, and 11. The result has prevailed that KNN is a promising tool to predict the output. The overall classification accuracy was 96.3%.

Figure 18 shows the confusion matrix for k-nearest neighbor with median average filter prediction and the figure shows that the maximum false positive value is 1, happening in class 3; whereas the maximum false negative is 1, which happened in class 1, 5, 6, 8, 10, and 11. The result prevailed that KNN is also a promising tool to predict the output with median average filter. The overall classification accuracy is 93.5%.

(B).
*Prediction via SVM*


Figure 19 shows the confusion matrix for the support vector machine with mean average filter prediction, and it shows that the maximum false positive value is 1, happening in class 2, 3, 6, 7, and 11; whereas the maximum false negative is 2, and this happened in class 6.

Figure 20 shows the confusion matrix for the support vector machine with median average filter prediction, and it shows that the maximum false positive value is 5, and this happened in class 2 and 3, whereas there are no false negative values. Even though the SVM accuracy is lower than the prediction accuracy of KNN, it predicts the output at 89.7% using mean average filter, and 82.4% using median average filter. The result shows that k-nearest neighbor predicts more false negative values than false positive values, whereas support vector machine predicts more false positive values than false negative values. Generally, the result prevailed that k-nearest neighbor is the best prediction tool with mean average filter.

## 5. Conclusions

This study focuses on the evaluation performance of a stratified TES tank installed at the GDC plant of UTP in terms of thermocline thickness, using machine-learning algorithms. The study was conducted using backpropagation with the Lavenberg–Marquadt algorithm, support vector machine, and k-nearest neighbor, trained with the temperature sensor data collected from the GDC plant installed at the university Teknologi PETRONAS, Malaysia. Based on the modeling results, the following conclusions can be drawn:The model was trained by using the data obtained from the temperature sensor data and non-linear regression model based on Musser and Behnfleth’s approach. The model was trained to predict $W_{TC}$ from temperature distribution data obtained from an operating TES tank. The average error during charging between ANN and actual values for $W_{TC}$ was 2.4%. The maximum R-square was 0.92.The ANN model for the TES tank was built based on various parameters i.e., number of hidden layers, numbers of hidden neurons, and learning rate. All the parameters were decided based on a trial-and-error approach. After performing various trials, the optimum parameters were decided as 10 numbers of neurons, 0.01 learning rate, and 1 hidden layer. On the decided parameters, the model results were validated with the results obtained by the non-linear regression model used for data generation. After validation with the model, the R-square achieved was 0.94 between the ANN and the non-linear regression model.The SVM model was also trained with the same data, the R-square obtained using the SVM model was 89% and for KNN it was 96.3%.Finally, based on the results, KNN outperformed the other machine learning models used in this study. The author would like to extend the use of the machine learning model for other parameter predictions, such as figure of merit and for optimization of the operations in future.

## 6. Research Limitations and Future Work

Although predicting the performance of stratified thermal energy storage tanks using ANN, SVM, and KNN produced excellent results, there are still some limitations in the work, such as:Less number of observations in dataset.The modeling lacks physics understanding.

The above-mentioned limitations can be improved further. The author considers that the work can still be extended as follows:
Consider employing more sophisticated machine learning models, such as deep learning models, which have remained unexplored in this area.Using physics-informed neural networks, which can solve the system’s governing equations and lessen the problem of data scarcity in the research.Planning for maintenance can be added to the work.

## Figures and Tables

**Figure 1 sensors-22-07687-f001:**
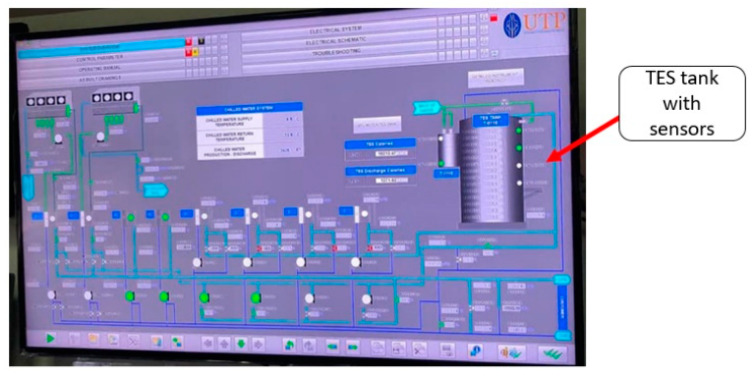
Photograph taken at GDC plant data acquisition.

**Figure 2 sensors-22-07687-f002:**
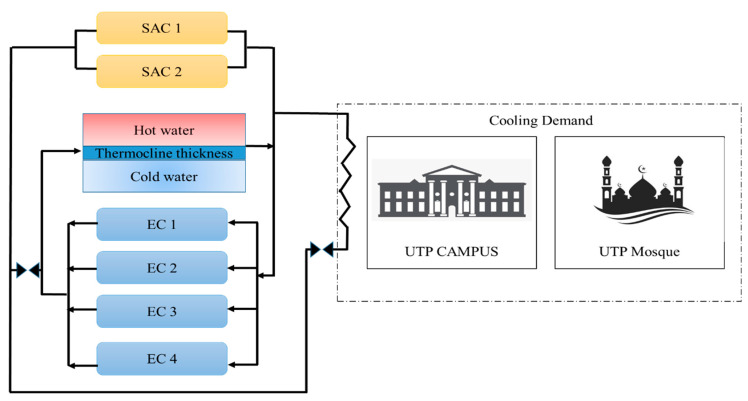
Flow diagram of charging cycle of TES at UTP GDC plant.

**Figure 3 sensors-22-07687-f003:**
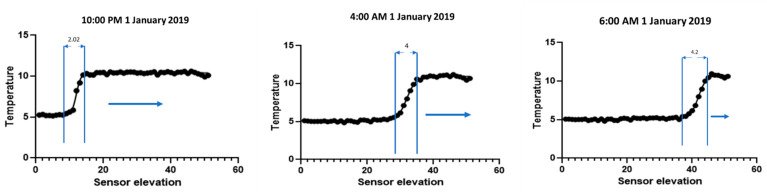
Evolution of temperature distribution and thermocline thickness during charging.

**Figure 4 sensors-22-07687-f004:**
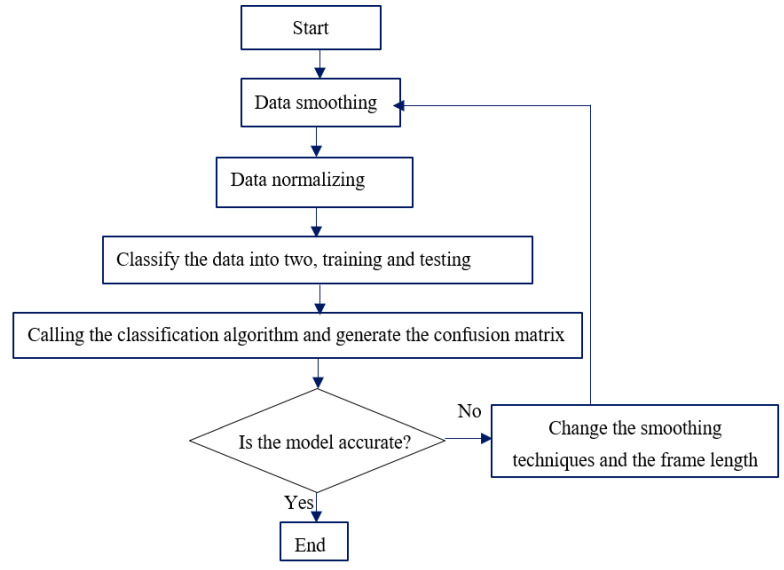
Prediction model development flowchart.

**Figure 5 sensors-22-07687-f005:**
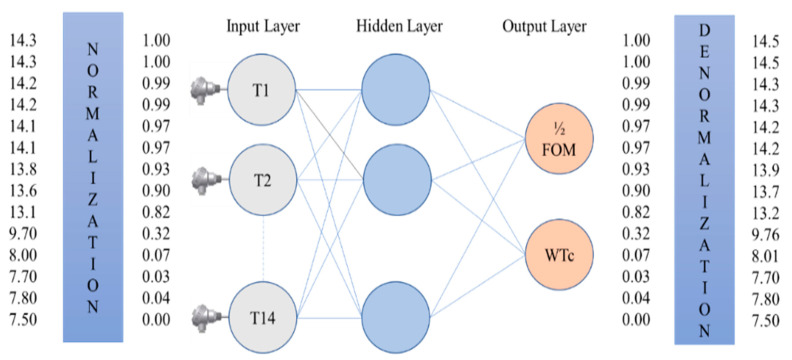
Example of data normalization.

**Figure 6 sensors-22-07687-f006:**
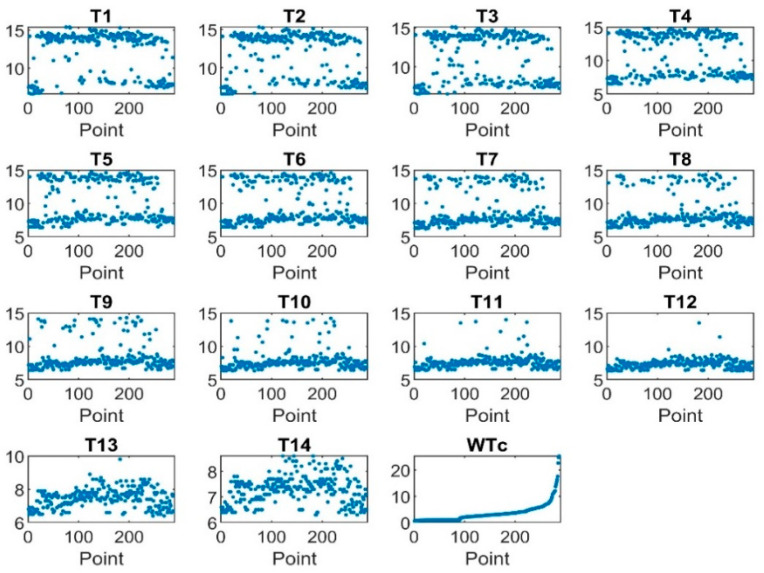
Input and output data without preprocessing.

**Figure 7 sensors-22-07687-f007:**
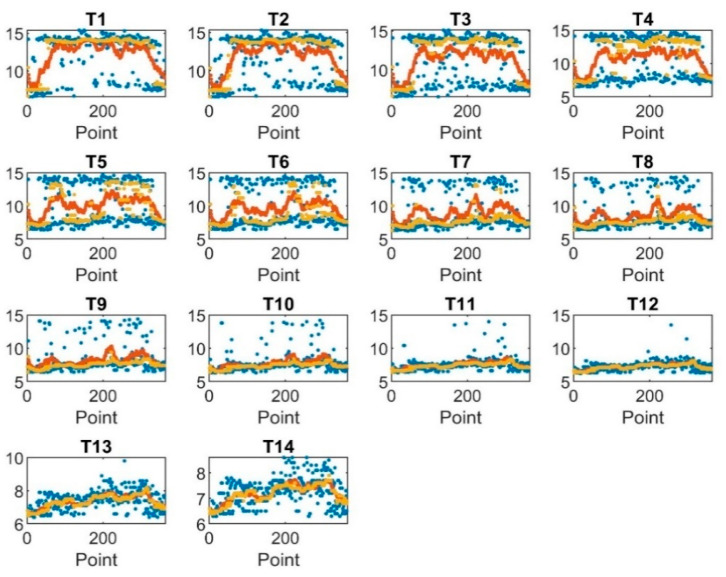
Smoothed versus unsmoothed data: blue color for unsmoothed data, red color smoothed data using mean average filter, and yellow for median filter.

**Figure 8 sensors-22-07687-f008:**
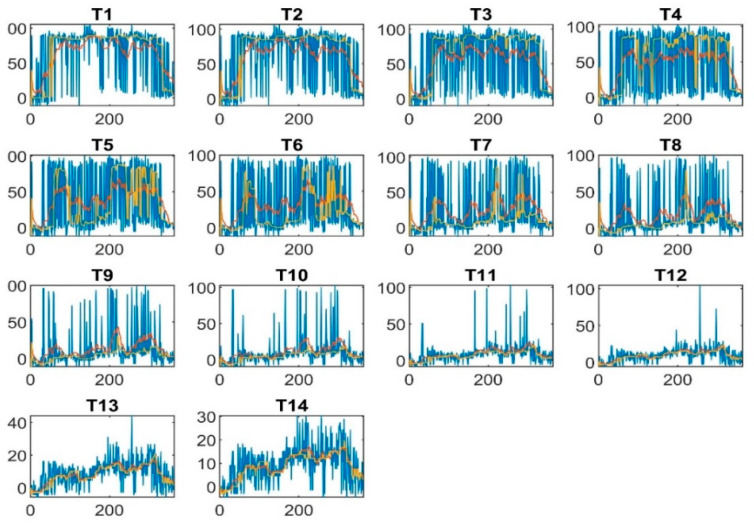
Smoothed and normalized versus unsmoothed and normalized data: blue for unsmoothed normalized data, red for mean average filter-smoothed and normalized data, and yellow for median average filter-smoothed and normalized data.

**Figure 9 sensors-22-07687-f009:**
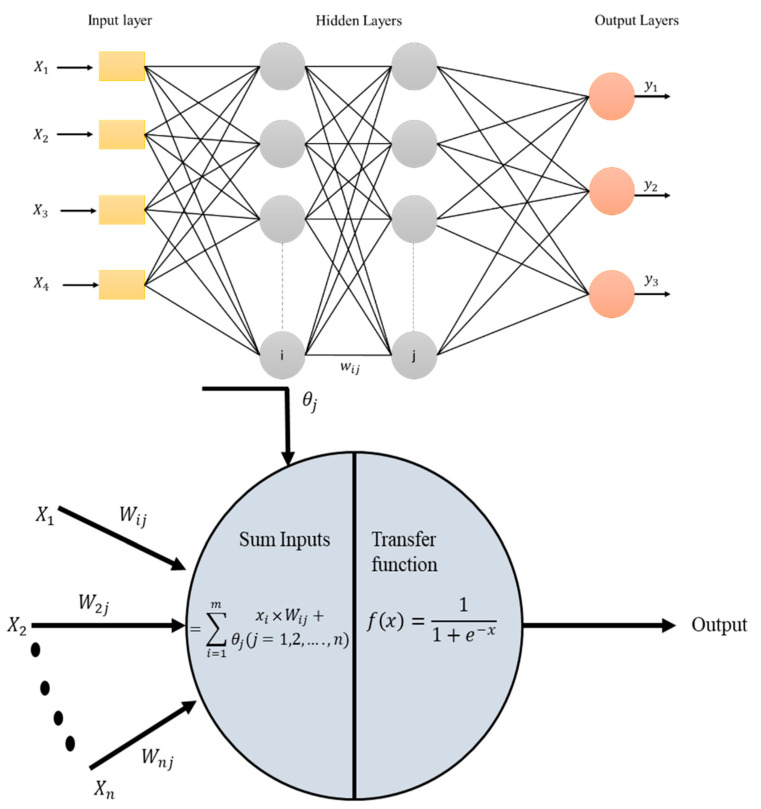
Basic architecture of artificial neural network and neuron.

**Figure 10 sensors-22-07687-f010:**
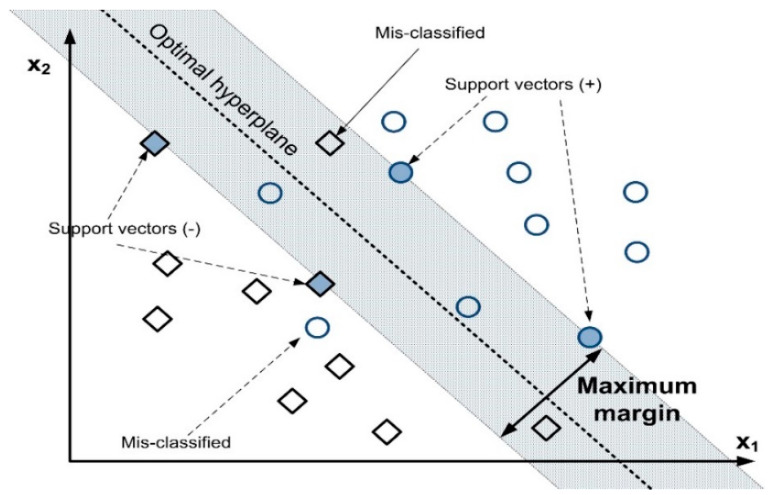
Concept of support vector machines [4].

**Figure 11 sensors-22-07687-f011:**
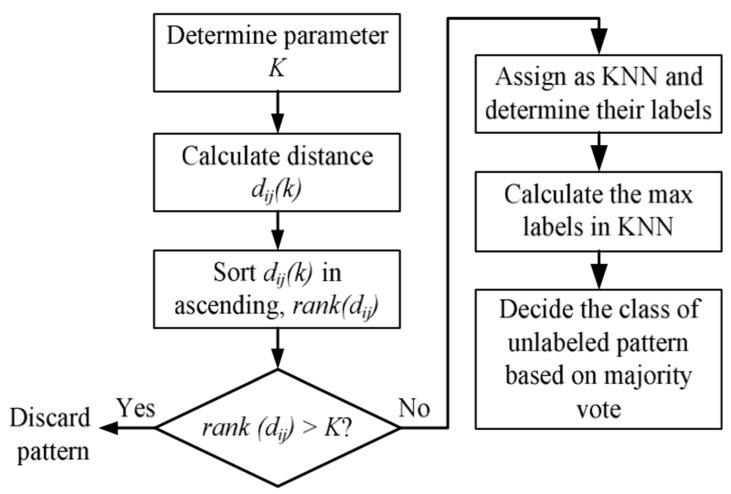
K-Nearest neighbor algorithm.

**Figure 12 sensors-22-07687-f012:**
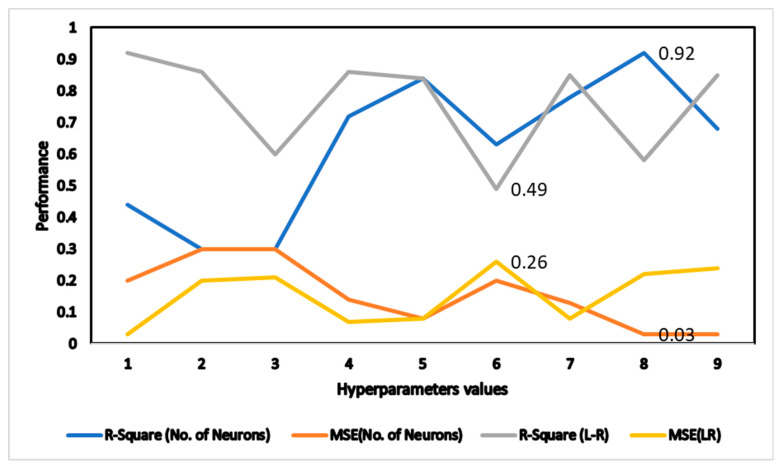
Impact of the number of neurons and learning rates.

**Figure 13 sensors-22-07687-f013:**
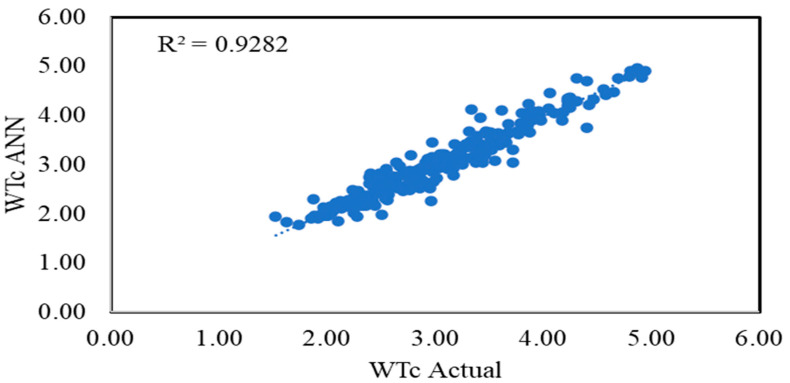
Regression of predicted and actual output values.

**Figure 14 sensors-22-07687-f014:**
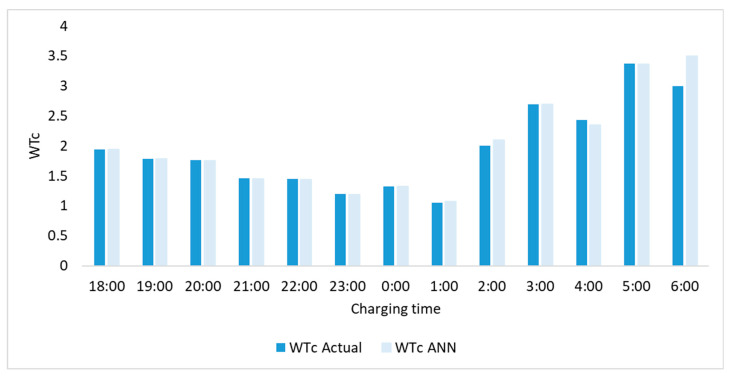
Comparison of output values.

**Figure 15 sensors-22-07687-f015:**
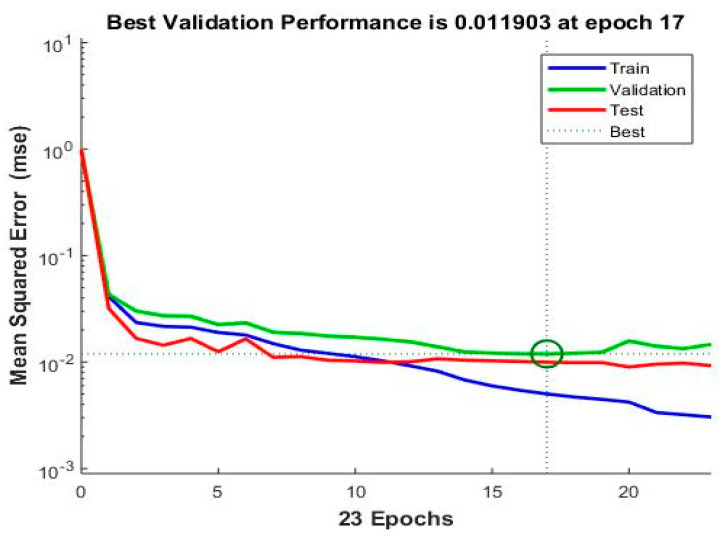
Performance of training, testing, and validation.

**Figure 16 sensors-22-07687-f016:**
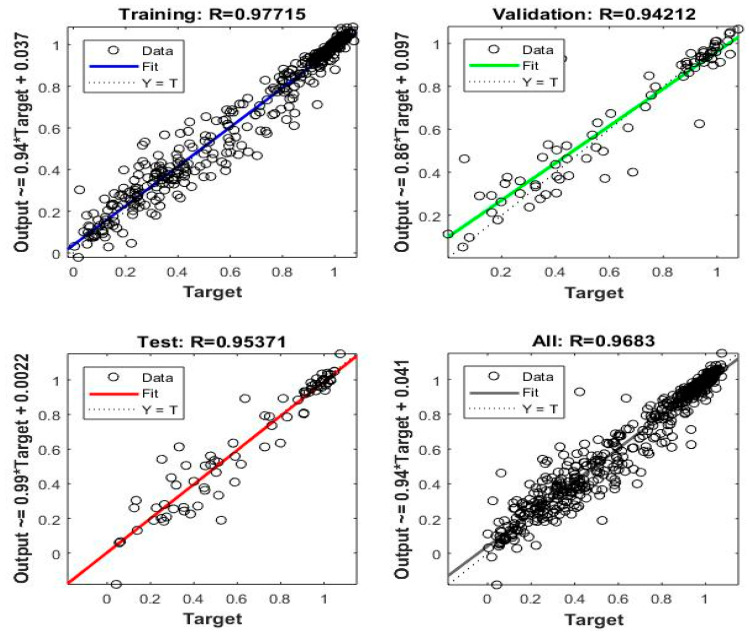
Regression analysis of training, validation, and testing.

**Figure 17 sensors-22-07687-f017:**
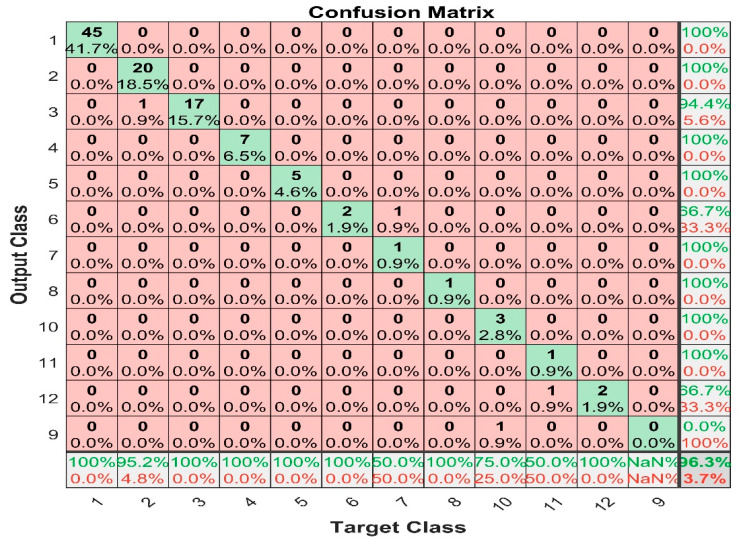
KNN with mean average filter confusion matrix.

**Figure 18 sensors-22-07687-f018:**
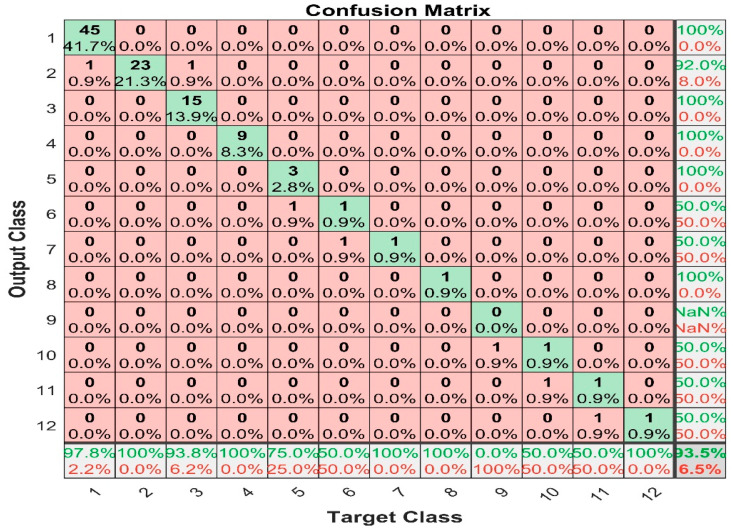
KNN with median filter confusion matrix.

**Figure 19 sensors-22-07687-f019:**
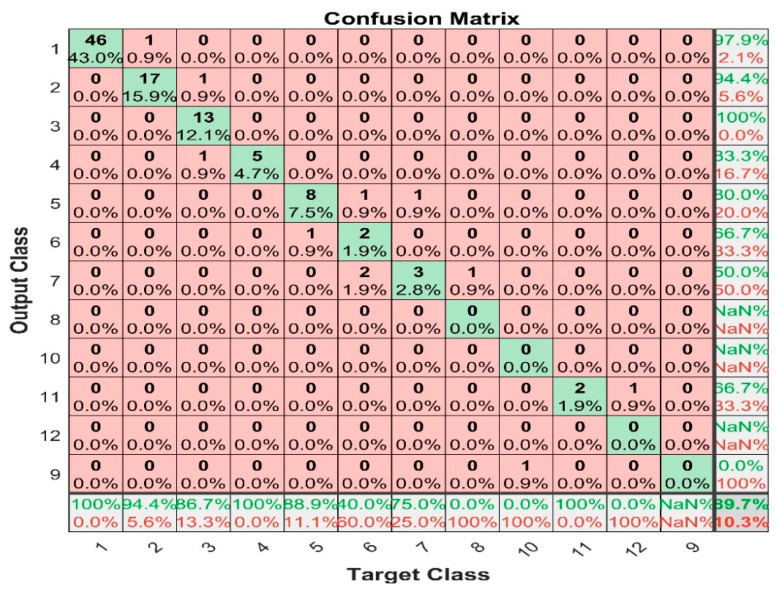
SVM with mean average filter confusion matrix.

**Figure 20 sensors-22-07687-f020:**
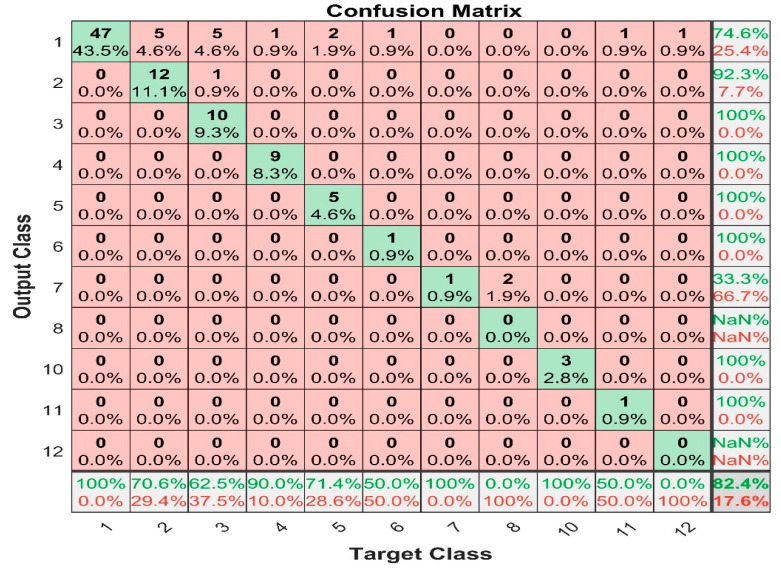
SVM with median filter confusion matrix.

**Table 1 sensors-22-07687-t001:** Example of division of dataset for one day data.

Division of Dataset	Hour	T1	T2	T3	T4	T5	T6	T7	T8	T9	T10	T11	T12	T13	T14	$W_{TC}$
70%	18:00	14.3	14.3	14.2	14.2	14.1	14.1	13.8	13.6	13.1	9.7	8.0	7.7	7.8	7.5	3.20
	19:00	14.3	14.2	14.2	14.1	14.1	13.9	13.5	13.0	9.4	8.0	8.0	7.7	7.9	7.6	3.18
	20:00	14.2	14.2	14.1	14.1	13.9	13.6	12.8	9.2	8.0	7.8	7.9	7.7	7.9	7.7	1.97
	21:00	14.2	14.1	14.1	13.9	13.5	12.8	8.8	7.9	7.9	7.8	7.9	7.7	7.9	7.8	2.26
	22:00	14.1	14.1	13.9	13.6	12.6	8.7	7.7	7.8	7.9	7.8	7.9	7.8	8.0	7.8	2.02
	23:00	14.1	13.9	13.5	12.5	8.3	7.9	7.7	7.8	7.9	7.8	7.9	7.8	8.0	7.9	2.03
	00:00	13.8	13.5	12.0	8.3	7.9	7.8	7.7	7.8	7.9	7.8	7.9	7.8	8.0	7.9	2.61
	00:01	13.4	11.5	8.0	7.9	7.8	7.8	7.7	7.8	7.9	7.8	7.9	7.8	8.0	7.9	2.26
	00:02	10.5	8.1	7.9	7.9	7.8	7.8	7.7	7.8	7.9	7.8	7.9	7.8	8.0	7.8	3.13
15%	00:03	7.9	7.9	7.9	7.9	7.8	7.8	7.7	7.8	7.9	7.8	7.8	7.5	7.7	7.3	3.04
	00:04	7.9	7.9	7.9	7.8	7.8	7.8	7.7	7.8	7.8	7.6	7.4	7.0	7.1	7.0	4.0
15%	00:05	7.9	7.9	7.8	7.8	7.8	7.8	7.7	7.8	7.5	7.0	6.9	6.7	6.8	6.7	33.1
	00:06	7.9	7.9	7.8	7.8	7.8	7.8	7.6	7.3	6.9	6.7	6.7	6.5	6.7	6.6	20.0

**Table 2 sensors-22-07687-t002:** Hyper parameters.

S.No	Model	Hyper Parameters
01	ANN	Hidden layers Hidden Neurons Activation function
02	SVM	Value of regularized terms C Kernel type Degree of kernel function Hyperplane
03	KNN	K—the number of neighbors

**Table 3 sensors-22-07687-t003:** Final Parameters for the model.

Sr No	Parameter	Remarks
01	Learning algorithm	LM
02	Activation function	Log-sigmoid
03	Number of neurons	10
04	Learning rate	0.01

**Table 4 sensors-22-07687-t004:** $W_{TC}$ magnitude and labels.

Input Data	$W_{TC}$ Magnitude	Class/Label
T_1_–T_14_	0.01–0.99	1
T_1_–T_14_	1.0–1.99	2
T_1_–T_14_	2.0–2.99	3
T_1_–T_14_	3.0–3.99	3
T_1_–T_14_	4.0–4.99	4
T_1_–T_14_	5.0–5.99	5
T_1_–T_14_	6.0–5.99	6
T_1_–T_14_	7.0–7.99	7
T_1_–T_14_	8.0–8.99	8
T_1_–T_14_	9.0–9.99	9
T_1_–T_14_	10.0–10.99	10
T_1_–T_14_	11.0–11.99	11
T_1_–T_14_	12.0–12.99	12

## Data Availability

Data available on request due to restrictions.

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
