# Peer review of "Machine Learning Approach to Predict the Performance of a Stratified Thermal Energy Storage Tank at a District Cooling Plant Using Sensor Data"

_sensors, 2022, doi:10.3390/s22197687_

Round 1

Reviewer 1 Report

The work focused on Machine Learning Approach to Predict Performance of Strati-2 fied Thermal Energy Storage Tank at District Cooling Plant Us-3 ing Sensor Data. The manuscript is within the scope of the Journal. However, the big issue is the original innovation of the article. Currently, there are a lot of papers focusing on ANN’s models. In order to help improve the paper quality, my suggestions and comments are shown below.

1) Abstract: abbreviation needs to be open. Not well prepared and the contribution is not straightforward.

2) Research gaps and originality need to be provided in introduction. The gaps need to be well summarised.

3) Fig. 9, why choose the transfer function as the one shown in the figure? How about other functions?

4) Please add more references in thermal energy storage, ANN and machine learning for dynamic performance prediction.

Sustainable Energy, Grids and Networks, Volume 21, March 2020, 100304

Renewable and Sustainable Energy Reviews, Volume 130, September 2020, 109889

Energy and Buildings, Volume 220, 1 August 2020, 110013

Building and Environment, Volume 174, May 2020, 106786

Renewable Energy, Volume 151, May 2020, Pages 403-418

Energy, Volume 202, 1 July 2020, 117747

5) Adding a new subsection on Research limitations and future study is necessary.

6) Comparison with related works from other researchers is necessary to highlight the significance of this study.

Overall, in terms of the journal of Sensors. The reviewers suggest the major revision.

Author Response

Response to Reviewer 1 Comments

Dear Reviwer, prior to responding, we are expressing our great thanks for handling, reviewing and providing valuable comments and suggestion, which with no doubt have enhanced and upgraded the article to a higher level. Your comments make the article much more beneficial to the thermal energy storage and machine learning related communities, whether researchers or planners and decision-makers. Thank you so much.

Response to Reviewer #1 Comments:

Point 1: Abstract: abbreviation needs to be open. Not well prepared and the contribution is not straightforward.

Response 1: The abbreviations has been opened. The contribution has been mentioned in the abstract.

Point 2: Research gaps and originality need to be provided in introduction. The gaps need to be well summarised.

Response 2: The gaps and originality has been added at the start and end of the introduction.

Point 3: Fig. 9, why choose the transfer function as the one shown in the figure? How about other functions?

Response 3: The transfer function i.e log sigmoid has been used in the ANN modelling due to its better prediction ability , the reference has been given.

Point 4: Please add more references in thermal energy storage, ANN and machine learning for dynamic performance prediction. Sustainable Energy, Grids and Networks, Volume 21, March 2020, 100304 , Renewable and Sustainable Energy Reviews, Volume 130, September 2020, 109889 , Energy and Buildings, Volume 220, 1 August 2020, 110013 , Building and Environment, Volume 174, May 2020, 106786 , Renewable Energy, Volume 151, May 2020, Pages 403-418

Energy, Volume 202, 1 July 2020, 117747

Response 4: As per suggestions more references have been added.

Point 5: Adding a new subsection on Research limitations and future study is necessary.

Response 4: The new subsection on Research limitions and future work has been added

Point 5: Comparison with related works from other researchers is necessary to highlight the significance of this study.

Response 5: The comparison with the non linear regression method has been already performed, which is validated with finite element analysis work in previous publicaitons.

We hope that the above explanation will satisfactorily answer the questions and concerns of the reviewers. We will be obliged if you would kindly accept the manuscript for publication.

We thank you for all your insightfull comments.

Authors,

Reviewer 2 Report

The paper outlines different AI and machine learning methods for sensing the temperature of a stratified TES tank. The methodology is well defined and systematically followed. I have some minor observations/comments to the authors:

1) Figure 12: define what is the x-axis and comment on the behavior of curves for R^2, etc. Legends need to be discriminated as a and b (missing from the graph) .

2) The authors in the results talked about figures 5 and 6 and then suddenly jumped to Figures 13 and 14. For ease of reading, the authors need to discuss the results with reference to the figures as they appear in the text.

3) Can the auths shed light on the total number of data points used in this study and how many were used for training, testing, and validation? 

 4) I'd like to know if KNN resulted in R^2 of around 0.96, why the authors are suggesting the use of deep machine learning? 

Author Response

Response to Reviewer 2 Comments

Dear Reviewer, prior to responding, we are expressing our great thanks for handling, reviewing, and providing valuable comments and suggestions, which without a doubt have enhanced and upgraded the article to a higher level. Your comments make the article much more beneficial to the thermal energy storage and machine learning-related communities, whether researchers or planners or decision-makers. Thank you so much.

Response to Reviewer #2 Comments:

Point 1: Figure 12: define what is the x-axis and comment on the behavior of curves for R^2, etc. Legends need to be discriminated as a and b (missing from the graph).

Response 1: In Figure. 12 X-axis and Y-axis have been defined. The comments on the behavior of R-Square and MSE have been discussed in the text. Refer to section 4.1

Point 2: The authors in the results talked about figures 5 and 6 and then suddenly jumped to Figures 13 and 14. For ease of reading, the authors need to discuss the results with reference to the figures as they appear in the text.

Response 2: The discussion has been re-arranged according to the numbering of the figures the in text.

Point 3: Can the auths shed light on the total number of data points used in this study and how many were used for training, testing, and validation? 

Response 3: The numbers of data points have already been discussed with the percentage of training, validation, and testing. For clarification, the text is highlighted in the abstract(line no. 22-23).

Point 4: I'd like to know if KNN resulted in R^2 of around 0.96, why the authors are suggesting the use of deep machine learning? 

Response 4: As discussed in the text no machine learning is proven to outperform others, that’s why we further proposed deep learning and physics-informed neural networks with more datasets to check the accuracy and applications. 

We hope that the above explanation will satisfactorily answer the questions and concerns of the reviewers. We will be obliged if you would kindly accept the manuscript for publication.

We thank you for all your insightful comments.

Authors,

Reviewer 3 Report

A good effort has been done by the authors to analyze the performance of 14 stratified thermal energy storage tank (TES) at Gas District Cooling plant (GDC) using sensor data. The performance of these TES has been measured using a variety of methodologies, both numerical and analytical. The hyperparameters for the three machine learning models, such as Artificial neural networks (ANN), support vector machines (SVM), and K-Nearest Neighbor (KNN), are chosen at optimal condition, and trial and error method has been used to select the best hyperparameters value.

The topic is interesting with many practically applications. However, before publication, the authors need to properly address the minor issues below:

1- In the Abstract, you should add main quantified results to this section.

2- The end of the introduction should be rewritten to better include novelty of your work.

3- In the Development of prediction model: Why the authors consider the three machine learning models, such as Artificial neural networks (ANN), support vector machines (SVM), and K-Nearest Neighbor (KNN)? What about the deep learning models to further enhance accuracy and computation speed.

 4- In the Nomenclature, the symbol (O-) hasn't a nomenclature.

Author Response

Response to Reviewer 3 Comments

Dear Reviewer, prior to responding, we are expressing our great thanks for handling, reviewing, and providing valuable comments and suggestions, which without a doubt have enhanced and upgraded the article to a higher level. Your comments make the article much more beneficial to the thermal energy storage and machine learning-related communities, whether researchers or planners, or decision-makers. Thank you so much.

Response to Reviewer #3 Comments:

Point 1: In the Abstract, you should add the main quantified results to this section.

Response 1 The main results have been added.

Point 2: The end of the introduction should be rewritten to better include the novelty of your work.

Response 2: The end of the introduction has been rewritten to the best of the author and the novelty has been clearly stated.

Point 3: In the Development of prediction model: Why do the authors consider the three machine learning models, such as Artificial neural networks (ANN), support vector machines (SVM), and K-Nearest Neighbor (KNN)? What about the deep learning models to further enhance accuracy and computation speed.

Response 3: Due to less number of dataset author does not consider deep learning as the best tool, it has been recommended for future work with a large dataset.

Point 4: In the Nomenclature, the symbol (O-) hasn't a nomenclature.

Response 4: In Nomenclature, the symbol (O) has been added.

We hope that the above explanation will satisfactorily answer the questions and concerns of the reviewers. We will be obliged if you would kindly accept the manuscript for publication.

We thank you for all your insightful comments.

Authors,

Round 2

Reviewer 1 Report

Accepted